# Estimation of Linkage Disequilibrium, Effective Population Size, and Genetic Parameters of Phenotypic Traits in Dabieshan Cattle

**DOI:** 10.3390/genes14010107

**Published:** 2022-12-29

**Authors:** Hai Jin, Shuanping Zhao, Yutang Jia, Lei Xu

**Affiliations:** Anhui Province Key Laboratory of Livestock and Poultry Product Safety Engineering, Institute of Animal Husbandry and Veterinary Medicine, Anhui Academy of Agricultural Sciences, Hefei 230031, China

**Keywords:** Dabieshan cattle, linkage disequilibrium, effective population size, heritability

## Abstract

Dabieshan cattle (DBSC) are a valuable genetic resource for indigenous cattle breeds in China. It is a small to medium-sized breed with slower growth, but with good meat quality and fat deposition. Genetic markers could be used for the estimation of population genetic structure and genetic parameters. In this work, we genotyped the DBSC breeding population (*n* = 235) with the GeneSeek Genomic Profiler (GGP) 100 k density genomic chip. Genotype data of 222 individuals and 81,579 SNPs were retained after quality control. The average minor allele frequency (MAF) was 0.20 and the average linkage disequilibrium (LD) level (r^2^) was 0.67 at a distance of 0–50 Kb. The estimated relationship coefficient and effective population size (Ne) were 0.023 and 86 for the current generation. In addition, we used genotype data to estimate the genetic parameters of the population’s phenotypic traits. Among them, height at hip cross (HHC) and shin circumference (SC) were rather high heritability traits, with heritability of 0.41 and 0.54, respectively. The results reflected the current cattle population’s extent of inbreeding and history. Through the principal breeding parameters, genomic breeding would significantly improve the genetic progress of breeding.

## 1. Introduction

Studying the level of genetic diversity within populations can help us understand population productivity, growth, and stability, as well as interspecific interactions within communities and ecosystem-level processes [1]. People are now paying greater attention to the protection of genetic resources.

China has plentiful ecosystems and abundant cattle resources, including 55 indigenous Chinese cattle breeds. They are reared in specific geographic regions, are widely distributed across China, and possess valuable genetic resources [2]. However, since the 1980s, a great number of exotic commercial cattle breeds have been introduced to blindly improve Chinese local cattle [3], resulting in a significant reduction in the quantity of local cattle breeds. Fewer purebred individuals of the local cattle breeds remain, leading to inbreeding, loss of genetic diversity, and even danger of extinction. It is urgent to protect the local variety of resources. Dabieshan cattle (DBSC) are one of the typical Chinese native cattle breeds, mainly distributed in the surrounding areas of the Dabie Mountains. They express a better performance in strong resistance to local diseases and parasites, low-quality feed (coarse fiber feed) resource tolerance, and high fat deposition capabilities [3]. The breed population head size is approximately 300,000–400,000 and mostly breeding scale in scatter-feed is 10–20 [3]. As such, DBSC generally have a small population size, and the traditional breeding methods have long generation times and lower effects. Now genomic selection (GS) is available to improve breeding efficiency and accelerate genetic progress [4] in animals and has been studied and successfully applied in the Holstein breeding populations, with the commercial chip containing around 50,000 SNP [5]. The developments of genome sequencing and SNP genotyping technologies, and new statistical tools have led to new technologies in evolutionary and developmental biology, and animal breeding. In developed countries, whole genome sequence analysis and GS are being applied in breeding schemes of beef cattle [6].

The estimation of the extent of the genome-wide linkage disequilibrium (LD) is vital for the number of markers that are needed in genome-wide association analysis (GWAS), GS, and effective population size (Ne). The availability of high-density single nucleotide polymorphism (SNP) genotyping platforms makes it possible to investigate LD better. The LD between molecular markers reflects the correlation between the genotypes of two markers or the degree of non-random association between their alleles [7]. The difference of allele frequency will affect the extent of LD, such as migration, mutation, selection, genetic drift, or other genetic variation that is experienced by the population [8]. As such, the LD can provide important information about population history and selection strength in a diverse population [9,10].

Understanding genetic diversity is essential for developing conservation programs in autochthonous breeds, and Ne is one of the most commonly used parameters to assess the loss of genetic diversity per generation and the increase in breeding per generation [11]. Ne refers to the ideal population content with the same gene frequency variance or the same heterozygosity attenuation rate as the actual population [12,13], which is an essential component of population genetics research. It determines the increment of the population average inbreeding coefficient and reflects the average homozygous rate of genes in the population genetic structure. The study of Ne is conducive to a clearer understanding of population evolution history and the genetic mechanism of complex traits. It also plays a critical indicator role in livestock and poultry breeding [14]. The population is very likely to be relatively closed when the effective population of a natural population is small, indicating that the population rarely experiences migration or is crossed with other herds or breeds and loses genetic variation quickly. At the same time, with a small Ne, it is difficult to avoid prematurity and high inbreeding levels, which will affect the heterogeneity and comprehensive application value in varieties. Therefore, breeding livestock and poultry should maintain a minimum Ne [15].

An estimation of heritability in populations depends on the partitioning of observed variation into unobserved genetic and environmental factors. Genetic markers can help to estimate heritability in a novel way [16]. Since quantitative traits are greatly affected by the environment, the heritability of a quantitative trait locus (QTL) helps judge the value of this QTL in breeding. QTLs with high heritability have a more significant effect on variety improvement and screening. Thus, we can estimate genetic parameters to improve the breeding efficiency and obtain faster genetic progress to carry out scientific breeding.

The main purpose of this work is to investigate allele frequency distribution and estimate the extent of LD level (r^2^), Ne, and the heritability of body measurement traits in DBSC using GGP 100 K Bovine chip. This information can help us to understand the evolutionary history of DBSC and provide reference parameters for the conservation, selection, and breeding of DBSC.

## 2. Materials and Methods

### 2.1. Animals and Genetic Data

There are about 20,000 DBSC in the area where the experiment was conducted, with herding and natural mating. A total of 235 DBSC (female = 199, male = 36, adult, 3–5-year-old) constructed a population, which were randomly selected from the breeding population of Taihu Jiuhong Agricultural Comprehensive Development Co., Ltd., Anhui (Taihu), China. The founders of the breeding population were bought from farmers living around Dabie Mountainous areas since 2014. Genomic DNA was extracted from blood samples using a TIANamp Blood DNA Kit (Tiangen Biotech Co., Ltd., Beijing, China). All the samples of 235 DBSC were genotyped with GGP Bovine 100 K (Neogen Inc., Lincoln, NE, USA). The principal component analysis (PCA) of the population was analyzed using Plink v1.09 to test the genetic background differences among individuals in the population.

Only SNPs that were located on 29 autosome chromosomes were considered for subsequent analyses. SNPs with (1) genotype call rate <90%, (2) MAF < 1%, (3) significant deviation from Hardy–Weinberg equilibrium (*p* < 10^−6^), and (4) no variation among the studied individuals were removed. Moreover, individuals with a missingness value >5% were excluded from further analysis. If a SNP pair had r^2^ > 0.998, only one SNP was kept for the next study pair. Initial data quality control of genotype data were utilized by Plink v1.09 [17] to exclude ineligible individuals and SNPs before data analysis. The missing genotype data of SNPs was filled by BEAGLE v4.1. Filtering was equal across chromosomes.

### 2.2. Minor Allele Frequency (MAF)

The MAF for all autosomal SNPs was estimated by Plink v1.09. The distribution of the allelic frequencies was analyzed using R and summarized as the proportion of the SNPs that were represented in five different categories of MAF: ≥0.05 to <0.1, ≥0.1 to <0.2, ≥0.2 to <0.3, ≥0.3 to <0.4, and ≥0.4 to <0.5. The results were graphed for comparison using R.

### 2.3. Phenotypic Data

The phenotypic data include adult body weight (BW) and body measurement traits. Body measurement traits include wither height (WH), height at hip cross (HHC), diagonal body length (DBL), chest girth (CG), abdomen circumference (AC), waist angle width (WAW), ischial end width (IEW), and shin circumference (SC). The descriptive statistical analysis of each trait was carried out to eliminate the abnormal value (except for three times of standard deviation) before the follow-up analysis. After collecting the original data, the phenotypes were corrected for fixed effects. Fixed effects in the model included gender, farm, and birth year. The significance of fixed effects is shown in the Appendix A.

### 2.4. Estimation of LD

The r^2^ statistic can be used to estimate the extent of LD. We calculated the r^2^ between SNP pairs with physical distances between 0 and 1 Mb of all autosomes to estimate the extent of LD in Plink v1.9. The decays of LD were analyzed for each of 2.5 kb between SNP pairs with an interval of less than 99,999 SNPs and 1 Mb, and plotted against the distance range. The r^2^ is the squared correlation of the alleles at two loci [18] and ranges between 0 and 1, which was estimated as follows:(1)$$r_{ij}^{2}=\frac{{\left(p_{ij}-p_{i}\times p_{j}\right)}^{2}}{p_{i}\left(1-p_{i}\right)\times p_{j}\left(1-p_{j}\right)}$$
where $p_{ij}$ is the haplotype frequency between the i marker and the j marker, $p_{i}$ and $p_{j}$ are the frequencies of the i marker, and the j marker, respectively. The average r^2^ within each marker interval is the arithmetic mean of r^2^ between all the markers within the interval.

### 2.5. Estimation of Ancestral and Contemporary Ne

Estimates of Ne were implemented in the GONE software [19], which utilizes genetic algorithms [20] in inferring demography history data and works with a small sample of individuals. This method is based on the relationship between the linkage disequilibrium that is observed between pairs of SNPs and Ne, and allows for nonlinear changes in Ne, such as population bottlenecks and expansions. The GONE program was run with default parameters, including an unknown phase, an average rate of recombination of 0.01 cm/Mb, and a genetic distance correction based on Haldane’s function. 

### 2.6. Estimation of Individual’s Genomic Relationship 

In this work, the genetic relationship matrix (GRM) between individuals represents the relationship between cattle, which is constructed by the genome array data and calculated with the following formula provided by VanRaden [21]:(2)$$G=\frac{Z\;Z\;\text{′}}{2\sum p_{i}\left(1-p_{i}\right)}$$
where Z is a matrix of marker genotypes of all individuals; 0, 1, and 2 represent the genotypes AA, AB, and BB, respectively; $p_{i}$ is the minor allele frequency at locus i; snf $Z^{\prime }$ is the transpose matrix of Z. 

### 2.7. Estimation of Genetic Parameter

We used a restricted maximum likelihood (REML) method to calculate heritability with an animal model and accounted for relationships between animals by using a G matrix. The animal model included random additive polygenic effects, fixed effects, and residuals for all traits. The additive polygenic effects were treated as random and assumed to be mutually independent. 

## 3. Results

### 3.1. Genotype Data and Quality Control

The GGP 100 K contains a total of 90,349 SNPs from 235 DBSC, and 81,579 SNPs from 222 DBSC with an average distance of 30.51 kb distributed over 29 chromosomes after quality control. The highest number of SNPs (5082 SNPs) was found on chromosome 1, and the lowest (1523 SNPs) was found on chromosome 25 ( Appendix A). The length of each chromosome, number, percentage of SNPs, and the average interval between SNPs for each chromosome are shown in Appendix A. The distribution of SNPs on each chromosome before and after quality control suggested that the number of unqualified SNPs accounts for the equivalent proportion on each chromosome.

### 3.2. MAF, GRM, and PCA

The distribution of MAF of all SNP markers (81,579 SNPs) on autosomes of DBSC are illustrated in Figure 1. The number of SNPs decreased with the increase in MAF frequency. The mean MAF across all autosomes was 0.20. The highest proportion of MAF was 31.17% between 0.05 and 0.1, and the lowest was 11.81% between 0.4 and 0.5. 68.83% of the SNPs show a MAF higher than 0.20.

The PCA showed that the first two principal components (PC1 and PC2) contributed 3.39% of the marker variation. Most of the population is clustered into one group and some individuals have high dispersion as displayed in Figure 2. 

The average coefficient of kinship according to the genomic relationship of the whole population was 0.023 by calculating the GRM.

### 3.3. LD and Ne

SNP pairs with an interval of 0–500 kb were selected from the results, and the decay of LD (r^2^ was calculated for every 2.5 kb) is shown in Figure 3. The LD level of SNPs decreased with the increase of the distance between SNP pairs on DBSC autosomes, and a sharp decay was observed from 0 to 100 Kb (details in Appendix A). The LD decay started when the level of average r^2^ was 0.70 at the bin of 0–2.5 kb and reached average r^2^ values of 0.29 for 400–500 kb. The average r^2^ was 0.50, 0.66, 0.46, 0.37, and 0.29 at an inter-marker distance of 2.5 kb, 30 kb, 50 kb, 100 kb, and 500 kb, respectively. The details of markers and LD (r^2^) between adjacent markers across autosomes are shown in Appendix A.

Demographic trajectories of up to 200th generations ago for DBSC were inferred in Figure 4. The pattern of the Ne curve was non-linear and kept fluctuating with up and down trajectories from 1064 (200th generation ago) to 86 (one generation ago). Also, questionable estimates were observed with a meteoric descent from 2051 (123th generation ago) to 1256 (119th generation ago), and a meteoric rise from 1110 (108th generation ago) to 2035 (97th generation ago).

### 3.4. Estimation of Genetic Parameters of Phenotypic Traits

The coefficient of variation of each trait ranged from 6.04% to 19.83% in the population (Appendix A). The higher coefficients of variation were IEW and BW with 19.83% and 18.65%, respectively. WH, HHC, and DBL had lower coefficients of variation, with respective values of 6.36%, 6.04%, and 8.49%. 

The estimation of additive variance ($\sigma_{a}^{2}$), residual variance ($\sigma_{e}^{2})$, and heritability (h^2^) of each phenotypic trait are shown in Table 1. We found that IEW, AC, and WAW were low heritable traits, and heritability was 0.12, 0.14, and 0.19, respectively. WH, DBL, CG, and BW were medium heritable traits, with heritability of 0.28, 0.28, 0.3, and 0.37, respectively. HHC and SC were high heritable traits, with a heritability of 0.41 and 0.54, respectively.

## 4. Discussion

DBSC is one of the indigenous cattle breeds in China, distributed in Dabie Mountainous areas with smaller body sizes for climbing slopes. They exhibit abundant genetic diversity [3] and have overall development and utilization prospects. However, the number of DBSC has decreased significantly as the promotion with improved breeds of cattle could lead to the threat of loss of excellent genes and degradation of performance year by year. The final purpose of the breeding of DBSC is to improve meat performance, such as increasing meat production, growth rate, and others.

### 4.1. MAF, GRM, and PCA

In this work, we found that the 29 autosomes of DBSC had 81,579 SNPs when using GGP 100 K for genotyping after quality control. The average MAF was 0.20, and an immense proportion of SNPs in the low MAF categories (0.05–0.2) was found. A previous study revealed that the average MAF of Chinese indigenous cattle and Simmental cattle were 0.24–0.27 and 0.29, higher than the DBSC in this study [4], indicating that the number of polymorphic loci of allele frequency is lower in DBSC. Based on the mitochondrial DNA analysis and Y-SNPs and Y-STRs markers tests, DBSC proved to only have *Bos indicus* paternal origin [22]. O’Brien et al. [10] compared the average distribution of MAF for *Bos indicus* breeds and *Bos taurus* breeds and demonstrated a general tendency in the indicine breeds to have more SNPs with lower MAF showing less than 0.2. Some researchers attribute this to the SNPs that were used in the assay were detected in European *Bos taurus* breeds, resulting in the polymorphism higher in *Bos taurus* and lower in *Bos indicus* [23]. However, we consider the proportion of lower MAF in *Bos indicus* may be greater in comparison to *Bos taurue*, which seems to be a characteristic for the species with lower genetic diversity assessed from sequence data.

The GRM was calculated based on the genome relationship G matrix construction method that was proposed by Van Raden [21]. Compared with the relationship that was calculated based on pedigree information, GRM could be more accurate in assessing the genetic relationship between siblings when the marker density is appropriate (at least 2500 SNPs [24]). The average coefficient of kinship over all 235 animals was 0.023 according to GRM. The result was lower than the corresponding estimates of 0.0533 and 0.0575 that were reported for US and Canadian Holsteins in 2007 [25], and also lower than the grouped American Angus cattle population (0.038–0.188) evaluated by Saatchi et al. [26]. Thus, the population (*n* = 235) may have less degree of inbreeding.

Most individuals of the population are genetically similar, shown as a dense cloud in the PCA. Although some individuals genetically diverge, they have an extent of correlation with each other, indicating that they maybe derive from the same population. A large individual variation within the population or the introduction of genetic material of other breeds may be reasons to explain the genetic variation between some individuals in the population.

### 4.2. Extent of LD

Both r^2^ and D’ statistics can be used to estimate the extent of LD, but D’ is more sensitive to changes in the effective population size and gene frequency [27]. As such, we use r^2^ to estimate the extent of LD. This work uses the LD decay analysis up to 500 kb for SNP pairs. The average r^2^ was 0.50 at a distance of 0–50 kb. The extent of LD is distinct among different cattle breeds. At an inter-marker distance of 10 kb, the average r^2^ value was 0.53 for DBSC, higher than Angus (0.46) and Hereford (0.49) [8]. Lu et al. [28] found that the extent of LD of Angus, Charolais, and crossbred beef cattle were 0.29, 0.22, and 0.21, respectively, at the SNP marker distance of less than 30 kb. The estimate of LD for SNP pairs separated by 40–50 kb was higher in Simmental cattle (0.21) and Wagyu cattle (0.21) compared to Chinese cattle (0.14–0.20) [4]. We observed that the extent of LD is greater, and the LD decay is slower up to 500 k for DBSC than values from other cattle breeds found in the literature [4,8,28], which is likely related to a higher ancestral relatedness [29] or historically smaller effective population sizes [30], which may be caused by regional isolation such as from mountains. Moreover, the slower LD decay of DBSC had the characteristic trend of populations that has suffered a collapse in population size or a bottleneck [31]. 

The estimate of LD is also affected by chip density and sample size. Reports of SNPs with low allele frequencies tend to underestimate r^2^ of LD between markers. However, O’Brien et al. [10] concluded that unbiased estimates of LD were obtained provided that MAF > 0.05, unless low-density SNP coverage assays were used. Khatkar et al. [32] reported that a small sample size led to an overestimation of LD and also illustrated that the accuracy of r^2^ values can reach 0.85 when 55 samples were used for the calculation. According to this, the results of LD evaluation in this experiment were not affected because we used a 100 k chip, which belongs to a medium density chip, and the sample size of this test (*n* = 235) meets the requirements. 

### 4.3. Estimation of Ne

LD and Ne can be used to investigate the evolutionary history and genetic contents of population. With the development of molecular biology, SNP as an essential genetic marker has been used to estimate Ne [33]. Many LD-based software solutions fall short in addressing problems including bottlenecks, migrations, expansions, and drops [34]. Santiago et al. created GONE to provide a computational framework based on intricate theoretical and mathematical analyses to accurately estimate drastic changes in Ne and infer recent demography history [19]. Meanwhile, over relatively recent timespans of about 200 generations back in time, the method has been shown to be more accurate than other alternative coalescence and mutation-recombination-based methods [34]. Consequently, we decided to estimate Ne for DBSC up to 200 generations ago. The long-term selection of breeds in a particular direction will increase the proportion of dominant genotypes in the population, resulting in artificial high-intensity LD, decreasing the Ne. A relatively small Ne will also cause inbreeding, thus increasing the homozygous probability of harmful alleles and reducing individual adaptability. It dramatically impacts species’ genetic diversity and may eventually be on the verge of extinction. Therefore, it is crucial to know and control the reasonable Ne. In animal breeding, FAO recommends that the Ne of the population should be maintained at 50–100 to maintain an appropriate breeding plan [15]. In this experiment, the Ne of the one generation ago was 86, suggesting that the population has a size in the range of the FAO recommendation and with reasonable inbreeding control. It may owe to the fact that individuals of the group came from different regions, isolated by Dabie Mountainous areas, which further produced a strong isolation effect. Consequently, 235 DBSC in this test can be used as a population for further breeding.

### 4.4. Estimation of Genetic Parameters of Phenotypic Traits

In this work, we used REML with an animal model to estimate the genetic parameters of BW and body measurement traits for DBSC. The animal model can maximize the use of phenotype data and gain more accuracy in estimating genetic parameters. Heritability, defined as the proportion of phenotypic variation that is attributable to genetic variation, provides important information about the genetic basis of a trait [35]. Genotype data of SNPs can be used to construct the GRM between individuals. GRM can replace the genetic relationship matrix based on pedigree information when the pedigree records are incomplete [6]. REML can be used to accurately evaluate the genetic parameters of each trait of the population [36]. The heritability estimated by genotype data of SNP is generally lower than that estimated by pedigree records. As the SNP data cannot represent all SNPs in the whole genome, the genetic variance estimated by SNP data is also less than pedigree data. However, for traits with high heritability or with the increase in population size, the distance between genetic variance that is estimated by pedigree and genomic SNP data will decrease gradually [37,38].

In this work, the estimated heritability of BW for 235 DBSC by GGP 100 K was 0.37. The estimation of heritabilities could achieve different results using different population sizes and methods of the relationship matrix for the same traits. Saatchi et al. [39] estimated the heritability of birth weight, weaning weight, and the yearling weight of Simmental cattle by Bovine SNP50 BeadChip to be 0.40, 0.30, and 0.29, respectively. Gunia et al. [40] estimated the heritability of birth weight and weaning weight of 2682 Charolais cattle by high-density SNP panel (777 K SNP) to be 0.36 and 0.22. Moreover, our study also estimated the heritability of WH, HHC, DBL, CG, AC, WAW, IEW, and SC for DBSC. The heritability of HHC and SC of DBSC were 0.41 and 0.54, respectively, which are rather high heritability values. Therefore, increasing the selection intensity of traits of HHC and SC will play a more significant role in accelerating the population genetic breeding of DBSC. Unfortunately, the heritability of these traits based on genotype data of SNPs has few related references. However, our results can provide a relevant reference for research, which provides basic parameters for formulating subsequent breeding programs for the population.

## 5. Conclusions

In this experiment, we analyzed the genetic diversity of autosomes using the GGP 100 k chip, including the extent of LD, Ne, and heritability in a DBSC population (*n* = 235). We found GGP 100 K markers with abundant polymorphisms (81,579 SNPs for 29 autosomes) for DBSC. The population had a low extent of inbreeding. The extent of LD was great, and the decay of LD was slow at the marker distance of 0–500 kb, indicating that there might be a higher ancestral relatedness or historically smaller effective population sizes. The Ne decreased rapidly, meaning species conservation is needed. The phenotypic traits of HHC and SC could provide the references for selection for further breeding of DBSC.

## Figures and Tables

**Figure 1 genes-14-00107-f001:**
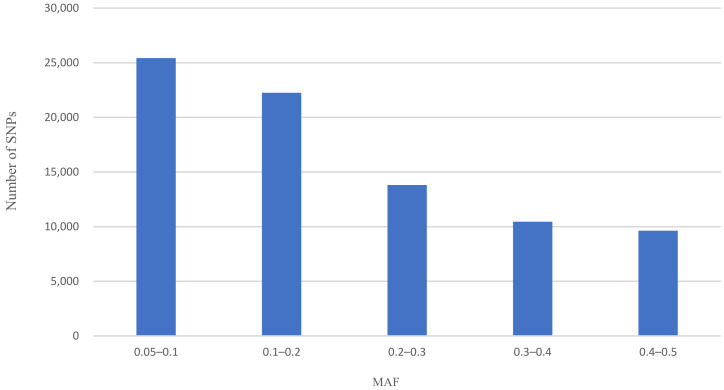
Distribution of minor allele frequency (MAF) for SNP after QC.

**Figure 2 genes-14-00107-f002:**
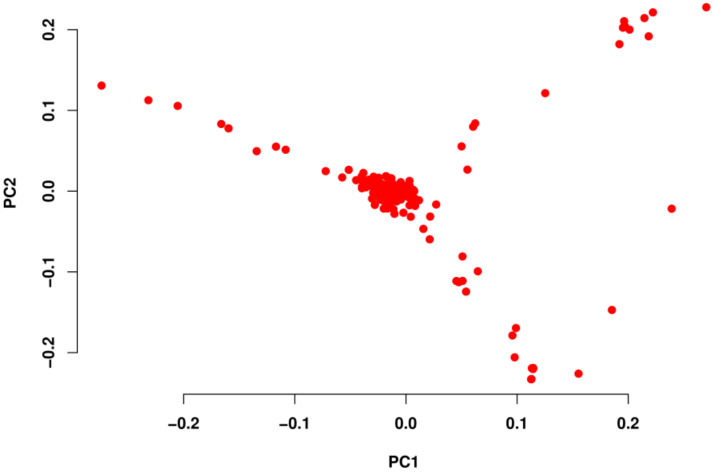
Principal component analysis (PCA) of the genomic relationship of the population of Dabieshan cattle (DBSC).

**Figure 3 genes-14-00107-f003:**
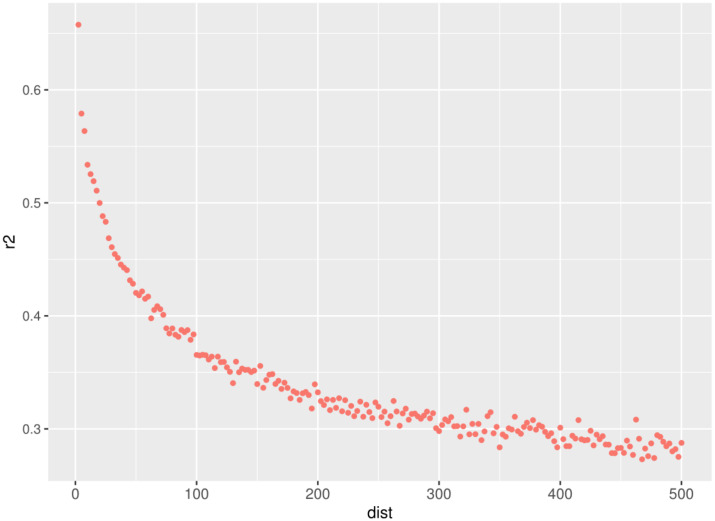
Linkage disequilibrium (LD) decay in different distances of the genome and up to 500 Kb on DBSC autosomes.

**Figure 4 genes-14-00107-f004:**
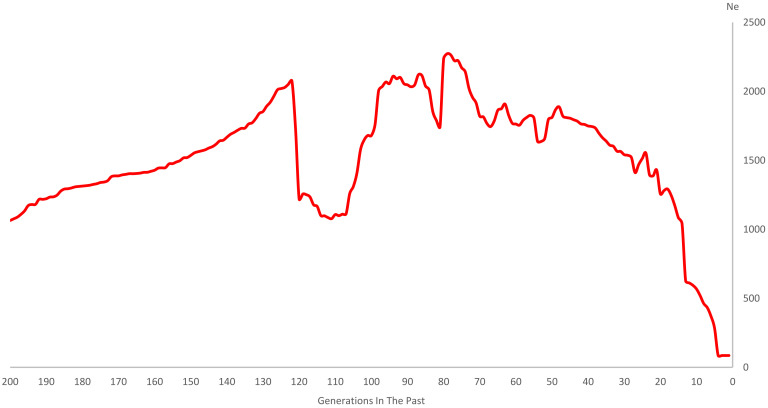
Genome-wide estimate of historical effective population size (Ne) over the past generations.

**Table 1 genes-14-00107-t001:** Genetic parameters of body size and weight traits from the population.

Phenotype Trait	$\sigma_{a}^{2}$	$\sigma_{e}^{2}$	h^2^ ± SE
Wither Height (WH)	16.523	42.488	0.28 ± 0.06
Height at Hip Cross (HHC)	22.189	31.930	0.41 ± 0.04
Diagonal Body Length (DBL)	36.585	94.075	0.28 ± 0.07
Chest Girth (CG)	133.103	310.574	0.30 ± 0.08
Abdomen Circumference (AC)	86.329	530.305	0.14 ± 0.06
Waist Angle Width (WAW)	5.049	21.525	0.19 ± 0.05
Ischial End Width (IEW)	1.523	11.167	0.12 ± 0.09
Shin Circumference (SC)	2.138	1.821	0.54 ± 0.07
Body Weight (BW)	1132.382	1928.109	0.37 ± 0.10

## Data Availability

All the data supporting the results of this study are included in the article and in the Additional File.

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
