# Peer review of "Estimation of Linkage Disequilibrium, Effective Population Size, and Genetic Parameters of Phenotypic Traits in Dabieshan Cattle"

_genes, 2022, doi:10.3390/genes14010107_

Round 1
Reviewer 1 Report
Jin et al. report about genomic characteristics of a rare Chinese breed. They present a nice dataset. However, there is improvement in the language, methods and discussion. The meaning of the sentences are often unclear to me. However, I am not a native speaker and thus I comment only on few mistakes where I do not get the message of the sentence.
Genetic diversity in terms of heterozygosity should be given in the results and compared to other breeds in the discussion. Otherwise the statement that they have high genetic diversity cannot be drawn from the study.
The filter steps for and after quality control can go in the supplementary material and just shortly stated in the text that the filtering was equal across chromosomes.
GRM does not necessarily correlate well with the true inbreeding level (Pong-Wong et al. 2021, The value of genomic relationship matrices to estimate levels of inbreeding). Thus I would recommend to estimate as well other inbreeding measures such as runs of homozygosity to get insight into the level of inbreeding in that breed.
Similarly estimates of Ne are very variable. Thus an additional estimate of contemporary Ne with an established software is recommended.
Also a kind of genetic structure analysis (e.g. STRUCTURE) should complement the PCA analysis.
Specific comments:
Introduction:
33 and 35: Write in relation to which breed DBSC has better fat deposition, slower growth rate and lower beef production.
38: This sentence is unclear. Do you mean reduction of inbreeding or increase of genetic diversity?
41: The meaning of “breeding methods have a long time” is unclear. Is long generation time meant?
43: What is more frequently used? The commercial chip?
55ff: Ne is not only useful to assess genetic diversity, but rather to estimate loss of genetic diversity per generation and increase in inbreeding per generation
65: what is meant with slight genetic differentiation?
75: remove “significantly”. What is meant with scientific breeding?
84: More information about the DBSC cattle breed would be helpful e.g. how many breeding populations are there, what’s the size of the total breeding population, how old is the breeding population and where do the founders come from. In the discussion is written that individuals come from different valleys. More information about the composition of the breeding population would be helpful.
Methods:
83: Include genetic data in the title. Maybe: “Animals and genetic data”
84: How were the cattle selected from the breeding facility? Randomly?
92: “no informative missing on chromosomal loci” can be removed to make the sentence easier.
What’s the difference between genotyping rate and missingness values?
95/96: one should be disequilibrium. How were SNPs removed for linkage equilibrium?
128: Why did you not use an established software? Please give a reference to the equation?
Results:
Figure 4: Is it possible to draw confidence intervals?
177/178: The comparison to another cattle breed should go in the discussion.
197: one generation ago instead of “the first”
202/203: Why have traits with higher variation higher selection potential? Maybe there is naturally more variance in that parameters and it is not necessarily heritability.
Write the sentences differently: The highest coefficients of variation had IEW and BW …. The lowest coefficients of variation had …
207: “are” instead of “were”?
Discussion:
215: Please give a reference for the statement of unique genetic diversity. If the statement refers to results of this paper a comparison with other breeds is missing.
MAF, GRM, PCA
229: use another phrasing than “some people”
233: please give a reference
240-246: What is meant by good population consistency? Do you mean with kess kinship a low relatedness among individuals? How could that be drawn from the PCA?
What is meant by certain genetic diversity? Which measure of the results shows genetic diversity?
Are there explanations or at least speculations why some individuals are further away in the PCA than most of the DBSC individuals? It would be nice to compare the PCA with published results from other breeds with the same SNP chip. Could it be that other breeds brought into the DBSC cattle referring to the end of the arms of the pca or do they rather reflect the different regions individuals are coming from or something else? Maybe a Structure analysis together with a map were the individuals come from would be helpful.
LD
251: “is” instead of “was”
258: remove “through comparison” and write something like “is slower up to 500k for DBSC than values from other cattle found in the literature.”
259: correct to “which may be caused by”
261: Please write if there is anything known about an experienced bottleneck by the DBSC breed.
Ne
285: First it is crucial to know the Ne and then it can be controlled, if that is possible.
286/287: remove “heads” as it sounds like one can count the number of individuals to get the Ne value.
288: “one generation ago” instead of “first generation ago”
288/289: Not the population structure ist protected, but the population has a size in the range of the FAO recommendation.
291: Formulate better why it can be used for further breeding without introducing genetic material of other breeds.
Phenotypig traits
296/297: Write the sentence in present form (are instead of was) and give a reference.
297-299: Heritabiliy is the proportion of the variation in a given trait within a population that is not explained by the environment or random chance. Please clarify this. The sentence is unclear in respect to population and environment.
317: Clarify what you mean by “belong to high heritability trait” or just write that these are rather high heritability values. Give a reference if it is defined somewhere when a trait belongs to low, medium or high heritability.
Conclusions:
326: clarify what you mean by high consistency. For the statement of low inbreeding coefficient another inbreeding value such as ROH would need to be estimated. Clarify what you mean by proper genetic diversity. In the moment no direct genetic estimation of genetic diversity such as heterozygosity is given in the manuscript.
330/331: If the traits are suitable depends also on what the breed is bred for. A little more background would be nice what the goal of the breeding is e.g. maintain health, increase meat production or milk production.
Reviewer 2 Report
The methods used are not novel and innovative but the results give some new information about possibilities to use the microarray data in conservation breed. Such an approach can bring useful data for breeding selection and genetic resources conservation. Nevertheless, in my opinion, the manuscript concerns a significant shortcoming which results the data obtained that may not be reliable – the authors used 235 animals representing different genders, probably maintained at different environmental conditions (farm and year effects at statistic model). There is a lack of description of phenotypic traits methods, and phenotypic traits can be affected by gender factors. All these points should be clarified.
Line 84 – please show the gender structure of the analyzed population
Line 88 – please show the full information about Neogen company according to the Guide for Authors outlines
Line 105 – the software version should be shown
Lines 109-115 –Were the used phenotypic data assessed by the unified methods for all animals tested? Were the animals kept in the same environmental and feeding conditions? The same maintenance conditions and unified methodology for traits evaluation are critical to performing the GWAS study. Moreover, if you analyzed animals with different sex (the gender factor at the model) at which ontogenesis period the phenotypic traits were measured and was it related to gender? Please clarify.
Figure 1 – This Figure is redundant. Such information is only related to method reliability and is not important for purpose of your study. It can be shown in the Supplement.
Line 150 – Were the „gender, farm, and measurement year” factors significant?
Line 151 - During which period the test animals were bred? Please show the year's number.
Table 1 – Did the gender factor affect the given traits or not?
Reviewer 3 Report
Estimation of Linkage Disequilibrium, Effective Population Size, and Genetic Parameters of Phenotypic Traits in Dabieshan Cattle
Only Introduction part of the MS needs some scholarly support and references.
The discussion portion needs justification of few more related studies to be added.
Otherwise the paper is very well written and the topic is very important for the health of cattle industry in china.
I would like to accept the MS after these minor suggestions.
Thank you
Round 2
Reviewer 1 Report
Jim et al. revised the manuscript well by taking the comments into account. I still recommend to add other Ne estimations as estimations of Ne are not very reliable, even among different methods based on linkage disequilibrium. Since Ne is an important parameter the estimates should be reliable. If they are in a certain range of several different methods an Ne estimate/range of the breed/population is more reliable. I suggest to use the LD method of NeEstimator (Do, C., Waples, R. S., Peel, D., Macbeth, G. M., Tillett, B. J. & Ovenden, J. R. (2014) NeEstimator V2: re-implementation of software for the estimation of contemporary effective population size (Ne) from genetic data. Molecular Ecology Resources, 14(1), 209-214) and GONE (Santiago, E., Novo, I., Pardiñas, A. F. Saura, M., Wang, J., Caballero, A. (2020). Recent demographic history inferred by high-resolution analysis of linkage disequilibrium. Molecular Biology and Evolution 37: 3642–3653). Both are easy to implement(NeEstimator needs Fstat or genepop format and GONE plink format).
I have some minor comments below and added a few suggestions with comments to improve the englisch in the attached pdf document. However, I am not a native englisch speaker and thus I well may have overlooked many mistakes.
When Ne is estimated by additional methods and the minor comments are clarified I suggest the manuscript is ready for publication.
Introduction:
38-40: why increase in inbreeding? Immigration of less related cattle breeds results in less inbreeding. Maybe you mean that only fewer purebred individuals of the local cattle breeds remain, which then leads to inbreeding. Also genetic diversity should increase with introduction of other cattle breeds. However, locally adapted genetic diversity is likely to get lost when a large number of commercial cattle is introduced.
So state the problem of inbreeding, loss of genetic diversity that is private to the local cattle breed and the risk of extinction in a separate sentence.
78-79:
The sentence is unclear. The process of evolution needs to explained in another sentence. It is not clear to me what you mean with that. Maybe that with a lower Ne the importance of genetic drift is increasing over selection and thus evolution is less effective?
A closed population has no migration and crosses. Two suggestions:
…..when the effective population size of a natural population is small, indicating that the population rarely experiences migration or crosses with other herds or breeds and looses genetic variation quickly.
…. when the effective population size of a natural population is small.
Methods
102: Please add when or about how many generations ago the breeding population was founded in order to have an idea how long ago the mixing of the individuals of different farms started.
109-111: The filter “missingness values” should be stated in a separate sentence as individuals with a missingness value >5% were removed and not SNPs. It is unclear what you mean by informative missing on chromosomal loci. Do you mean that you kept only polymorphic loci and removed monomorphic loci? One suggestion, if you meant that you removed monomorphic loci:
SNPs with (1) genotype call rate <90%, (2) MAF <1%, (4) significant deviation from Hardy-Weinberg equilibrium (p<10-6) and (5) no variation among the studied individuals were removed. Moreover, individuals with a missingness value >5% were excluded from further analysis.
112: It is still unclear to me what you mean by “removing the effect of loci in linkage equilibrium”? It states that you remove every locus, those with strong linkage disequilibrium and those in linkage equilibrium. Please clarify what you mean by that or remove it when you just removed SNPs in high LD. If you only remove loci that were also physically linked in addition to the high r2 value please write it as well.

Reviewer 2 Report
The authors have responded to all of the comments and suggestions and the manuscript has been improved significantly. I recommend moderate English editing.
Author Response
We edited English moderately. Please see the attachment.
